# Efficacy and Safety of Oral Administration of Wine Lees Extract (WLE)-Derived Ceramides and Glucosylceramides in Enhancing Skin Barrier Function: A Randomized, Double-Blind, Placebo-Controlled Study

**DOI:** 10.3390/nu16132100

**Published:** 2024-07-01

**Authors:** Angga Sanjaya, Akiko Ishida, Xuan Li, Yugweng Kim, Hiroaki Yamada, Takashi Kometani, Yusuke Yamashita, Young-il Kim

**Affiliations:** Pharma Foods International Co., Ltd., Kyoto 615-8245, Japan; a-ishida@pharmafoods.co.jp (A.I.); xuan-li@pharmafoods.co.jp (X.L.); y-kim@pharmafoods.co.jp (Y.K.); h-yamada@pharmafoods.co.jp (H.Y.); takashi-kometani@pharmafoods.co.jp (T.K.); y-yamashita@pharmafoods.co.jp (Y.Y.)

**Keywords:** ceramides, clinical study, glucosylceramides, skin barrier, transepidermal water loss (TEWL), wine lees

## Abstract

Background: Our search for plant-derived ceramides from sustainable sources led to the discovery of ceramides and glucosylceramides in wine lees. Objective: This study evaluated the efficacy and safety of wine lees extract (WLE)-derived ceramides and glucosylceramides in enhancing skin barrier function. Methods: A randomized, double-blind, placebo-controlled study was conducted with 30 healthy Japanese subjects aged 20–64. Subjects were allocated to receive either the WLE-derived ceramides and glucosylceramides (test group) or placebo for 12 weeks. The primary outcome was transepidermal water loss (TEWL), and secondary outcomes included skin hydration, visual analog scale (VAS) of itching sensation, and the Japanese Skindex-29. Results: One participant withdrew for personal reasons, resulting in 29 subjects for data analysis (placebo *n* = 15; test *n* = 14). The test group showed a tendency of lower TEWL compared to the placebo after 8 weeks (*p* = 0.07). Furthermore, after 12 weeks of administration, the test group had significantly lower TEWL than the placebo (*p* = 0.04). On the other hand, no significant differences were observed in the secondary outcome parameters. No adverse events related to the supplements were reported. Conclusions: Oral supplementation of WLE-derived ceramides and glucosylceramides is a prominent and safe approach to enhancing skin barrier function and health. Trial registration: (UMIN000050422).

## 1. Introduction

As the largest organ of the body, the skin plays a foremost role in maintaining physiological balance and safeguarding against external threats. Skin consists of three layers: the epidermis, dermis, and subcutis [1]. The outermost layer of skin, known as the stratum corneum (SC), is made up of multiple layers of keratinized corneocytes embedded in a lipid matrix with an ordered lamellar structure [2]. Human SC lipids are composed of 50% ceramides (CERs), 25% cholesterol, and 15% free fatty acids (FFAs) [3]. The intercellular lipids in SC play an important role in regulating skin water content. Ceramides are the most important lipids that form the permeability barrier of the skin. Epidermal ceramides are composed of long-chain sphingoid bases linked with FFAs via amide bonds and are primarily synthesized in the stratum spinosum (SS) of the epidermis [4]. Alteration in skin lipid profiles is known to cause defects in the permeability barrier, resulting in increased transepidermal water loss (TEWL). Furthermore, reduced levels of epidermal ceramides and the consequently diminished skin barrier function are recognized in a wide range of skin conditions, such as dry skin, atopic dermatitis, psoriasis, and skin aging [5].

Sphingolipids are a family of compounds that have a sphingoid base (SBs) with an amide-linked fatty acid and a polar head group, such as phosphorylcholine (for sphingomyelin) or carbohydrate (for cerebrosides, gangliosides, and other complex glycolipids) [6]. The most abundant class of sphingolipids in plant tissue are mono-glucosylceramides, which are mostly characterized by a double bond at position 8 on the sphingoid residues and α-hydroxy FAs [7]. While glucosylceramides are found in various plants, such as rice, konjac, wheat, and pineapple [8,9,10,11], ceramides have also been identified in soy sauce lees and citrus peel [12,13]. In recent decades, scientists have investigated the beneficial effects of oral supplementation of ceramides to improve dry skin, skin condition, and skin-associated discomforts. Animal studies have elucidated the oral bioavailability of glucosylceramides and ceramides. Dietary ceramides and glucosylceramides can be absorbed as intact molecules or, after being hydrolyzed, as free sphingoid bases, which are then resynthesized into ceramides with endogenous non-hydroxy fatty acids [14,15].

Winemaking has been a significant contributor in the alcoholic beverages industry in the context of volume and economic impact, with an estimated global production of 244 million hectoliters in 2023 [16]. Meanwhile, the winemaking process generates huge quantities of by-products, including grape marc (62%), wine lees (14%), grape stalk (12%), and dewatered sludge (12%) [17]. Utilizing these by-products could have positive impacts on both the economy and the environment. Among these by-products, wine lees, a residue generated during fermentation that consists of yeasts and grape debris, remains the least explored [18]. While ceramides and glucosylceramides have been identified in other fermented food by-products [19,20], we successfully identified ceramides and glucosylceramides in wine lees in a preliminary study.

In the present study, we aimed to identify ceramide and glucosylceramide molecular species derived from wine lees and to evaluate their safety and efficacy on skin barrier function.

## 2. Materials and Methods

### 2.1. Wine Lees Extract (WLE)

50 g of dried red wine lees (Tartaros Gonzalos Castilos L., Alicante, Spain) was extracted with 10 volumes (*v*/*w*) of distilled water at 60 °C for 1 h. The insoluble part was separated by filtration (Qualitative filter paper No. 1, Whatman, Maidstone, UK), followed by further extraction using ethanol (Fujifilm Wako Pure Chemical, Osaka, Japan) at 40 °C for 6 h. Finally, ethanol was removed by rotary evaporation, yielding 4 g (8%) of WLE, which was then used in the in vitro study.

### 2.2. Identification of Ceramides and Glucosylceramides

Ceramides and glucosylceramides were isolated and purified from wine lees by using column chromatography. Briefly, the ethanolic extract of wine lees was suspended in aqueous methanol and partitioned with n-hexane to yield the n-hexane fraction and aqueous methanol fraction. Subsequently, the n-hexane fraction was subjected to silica gel chromatography (Wakogel C-200, FUJIFILM Wako Pure Chemical, Osaka, Japan) and eluted with chloroform–methanol in a stepwise manner. Further purification was performed using silica gel 60 (FUJIFILM Wako Pure Chemical, Osaka, Japan) to obtain purified ceramides and glucosylceramides. Each step of column chromatography was monitored using TLC and HPLC. The isolated ceramides and glucosylceramides were compared with standard *Citrus unshiu* ceramide and konjac glucosylceramides (Nagara Science, Gifu, Japan). The image of the TLC plate is provided in Appendix A. Furthermore, the purified ceramides and glucosylceramides were then characterized using normal phase HPLC (Inertsil SIL 100-5, 4.6 mm i.d. ×250 mm, GL Science Inc., Tokyo, Japan) with an evaporative light-scattering detector (ELSD). Mobile phases were chloroform (A) and 95% methanol (B). The gradient profile was 1 to 25% B from 0 to 15 min, 25 to 90% B from 15 to 20 min, 90 to 100% B from 20 to 25 min, and 100% B to 0% A from 25 to 30 min at a flow rate of 1 mL/min. ELSD was performed at gain setting 6 with nitrogen gas pressure set at 350 kPa. The identification of ceramides and glucosylceramides molecular species was carried out by using liquid chromatography–mass spectrometry (LC-MS/MS) (Shimadzu 8045, Shimadzu, Kyoto, Japan) with an electrospray ionization (ESI) source. The ion spray voltage was maintained at 4.0 kV. The source temperature was set at 300 °C, and the nanoflow gas pressure was 230 kPa. The mass spectrum was acquired from *m*/*z* 100 to 1000 in the positive ion mode with an acquisition time of 1.5 min; the scan duration was 1 s. Methanol containing 0.1% formic acid (*v*/*v*) was used as the mobile phase at a flow rate of 0.1 mL/min. Mass spectra were obtained at 30–40 eV collision energy.

### 2.3. Analysis of Genes Expression in Human Epidermal Keratinocytes (NHEKs)

Normal human epidermal keratinocytes (NHEKs) from pooled adult donors (PromoCell, Heidelberg, Germany) were grown to 80% confluency and sub-cultured in keratinocytes growth medium 2 (PromoCell, Heidelberg, Germany). Early-passage cells (passage number 3–5) were seeded into a 24-well plate at a density of 5 × 10^4^ cells per well and cultured for 24 h. The cells were then incubated with or without WLE in keratinocyte basal medium 2 supplemented with 1.2 mM [Ca^2+^] for 72 h. After incubation, total RNA was extracted using Isogen (Nippon Gene, Tokyo, Japan) following the manufacturer’s instructions. The extracted RNA was reverse transcribed into cDNA using a cDNA reverse transcription kit (TAKARA-bio, Shiga, Japan). The RNA expression level was quantified by quantitative real-time PCR using TB green premix (TAKARA-bio, Shiga, Japan) and a thermal cycler dice real-time system (TAKARA-bio, Japan). The genes of interest include serine palmitoyltransferase long chain base subunit 2 (*SPTLC 2*), ceramide synthase 1 (*CerS 1*), ceramide synthase 3 (*CerS 3*), profilaggrin (*PFR*), involucrin (*INV*), and loricrin (*LOR*). Glyceraldehyde 3-phosphate dehydrogenase (*GAPDH*) was used as a reference gene. The primer sequences are provided in Appendix A.

### 2.4. Culture of 3D Human Epidermal Equivalents (3DHEEs)

3DHEEs (LabCyte^®^ EPI-MODEL 6D, J-TEC, Aichi, Japan) were conditioned for 24 h (37 °C, 5% CO_2_) prior to the addition of samples, as per the manufacturer’s instructions. WLE was dissolved in dimethyl sulfoxide (DMSO) and diluted 1000 times in the culture medium to a final concentration of 5 and 10 µg/mL. The mixtures of ceramides and glucosylceramides isolated from wine lees were dissolved in a similar manner to a final concentration of 0.42 and 0.82 µg/mL, which were equivalent to 5 and 10 µg/mL of WLE. Culture medium supplemented with 0.1% DMSO was used as a control. The 3DHEEs were then incubated for 5 days with daily medium changes. Finally, after medium removal, the epidermis was washed with 1 mL phosphate-buffered saline (PBS) three times and then stored at −80 °C until further analysis.

### 2.5. Quantification of the Ceramide Content in 3DHEEs

Total ceramides in the 3DHEEs were extracted using the Bligh and Dyer method with slight modifications [21]. Briefly, the epidermis recovered from the 3DHEEs was lysed by sonication (40 °C, 10 min) in 2 mL of chloroform–methanol (2:1, *v*/*v*). The crude extract was dried with nitrogen gas and redissolved in 100 µL of chloroform–methanol (2:1, *v*/*v*). The high-performance thin-layer chromatography (HPTLC) silica gel plate (Merck, Darmstadt, Germany) was developed in a saturated TLC chamber twice with a mixture of chloroform–methanol–acetic acid (190:9:1, *v*/*v*). The dried plate was visualized using a 10% copper sulfate and 8% phosphoric acid aqueous solution and heated to 180 °C for 8 min. Photographs of the developed thin-layer plate were taken using an imaging system (ATTO luminograph, Tokyo, Japan). Ceramide non-hydroxy-sphingosine (NS), non-hydroxy-phytosphingosine (NP), and α-hydroxy-phytosphingosine (AP) (Avanti Polar Lipids, Alabaster, Birmingham, AL, USA) were used as standards. The quantitative analysis of ceramides (ceramide NS, NP, and AP) was performed by densitometry.

### 2.6. Preparation of WLE for the Clinical Study

The test supplement used in the clinical study was manufactured by Pharmafoods International Co., Ltd. (Kyoto, Japan) (commercially available as CERAMINOL^®^). The test supplement consisted of mixtures of WLE and cyclodextrin (Cyclochem, Kobe, Japan). The daily dose of the test supplement (100 mg/day) contained ≥2 mg WLE-derived ceramides and glucosylceramides, which were verified by HPLC (Appendix A). Cyclodextrin with a color additive was used as placebo, ensuring that the test supplement and placebo were indistinguishable in appearance. The test supplement and placebo were encapsulated in cellulose-based capsules (100 mg/capsule) to facilitate oral administration. The nutritional composition of the test supplement and the placebo is provided in Table 1. The compositional analysis (except for ceramide contents) was analyzed by the Japan Food Research Laboratories (Tokyo, Japan).

### 2.7. Study Design and Procedures

This randomized, double-blind, and placebo-controlled study was conducted between February 2023 and June 2023 at Pharma Foods International Co., Ltd. (Kyoto, Japan). The study was conducted according to the Declaration of Helsinki and ICH Good Clinical Practice principles. The study protocol was approved by the Yoga Allergy Clinic Clinical Research Ethics Review Committee (Tokyo, Japan; Ethical approval code: RD11010TF04). All participants were fully informed about the study and gave written informed consent before participation. This study was registered in the University Hospital Medical Information Network Clinical Trials Registry (UMIN-CTR) under registration number (UMIN000050422).

After screening participants based on the inclusion and exclusion criteria specified in the following section, the included participants were stratified based on gender, age, and baseline TEWL and randomly assigned to receive either the test supplement or placebo by using random numbers generated in Microsoft Excel (2022). The allocation sequence was concealed and blinded to the participants, investigators, caregivers, and outcome assessors until the end of the study. The participants were instructed to take one capsule orally after dinner every day and to maintain their habitual lifestyle. Throughout the study, participants were prohibited from initiating the consumption of any functional foods or nutritional supplements. Additionally, on the day prior to examination, participants were requested to abstain from consuming alcohol or engaging in strenuous physical activity. Furthermore, until completion of the intervention, any usage of medications, functional foods, or nutritional supplements was recorded on a daily basis. The assessment of efficacy was conducted at weeks 0, 4, 8, and 12. Meanwhile, safety parameters were examined at weeks 0, 4, 8, 12, and 16 (Figure 1). The primary outcome was TEWL, and the secondary outcomes included skin hydration, the Visual Analogue Scale (VAS) of itching sensation, and the Japanese version of Skindex-29. The safety parameters included hematological blood-chemical tests, urine tests, and physical and vital sign measurements. Adverse events were monitored through participant interviews conducted by the physician and through self-reports throughout the study. The safety endpoint was the occurrence of adverse events related to the intake of a test supplement or placebo.

### 2.8. Study Participants

The inclusion criteria for this study were healthy Japanese male and female adults aged 20–64 with self-reported dryness of skin in a questionnaire survey. The exclusion criteria included: (1) history of serious diseases such as brain diseases, malignant tumors, immune disorders, diabetes, kidney or heart diseases, renal disease, thyroid disease, adrenal disease, and other metabolic diseases; (2) receiving treatment for chronic conditions, e.g., arrhythmia, dyslipidemia, and hypertension, as well as severe skin diseases, e.g., atopy and dermatitis; (3) regularly using medications, including herbal remedies, and drugs/quasi-drugs with skin improvement-related claims within the last two weeks prior to the study; (4) unable to avoid deliberate sunlight exposure; (5) allergies to medications and/or the test supplement and/or placebo; (6) pregnancy or breastfeeding or planning pregnancy; (7) subjects judged unsuitable based on screening; (8) other conditions deemed unsuitable for participation by the physician. Based on the data from a preliminary experiment, aiming at 80% power to detect a 0.9 standard deviation difference in TEWL at a significance level of 5%, and considering a 12% dropout rate, it was calculated that approximately 30 subjects were required.

### 2.9. Measurement of TEWL and Water Content

TEWL on the right and left center forearm was measured five times from multiple sites within an area of 3 cm^2^ using a Tewameter TM300 (Courage-Khazaka, Koln, Germany). Each measurement lasted for >60 s, with stable readings being obtained between 5 s and 30 s. The mean of five repeated measures was taken. Water content at the same anatomical positions was measured with a Corneometer CM825^®^ (Courage-Khazaka, Germany). Similarly, five repeated measures were taken, and the mean was used for the statistical analysis. All measurements were performed after 15 min acclimatization in a temperature- and humidity-controlled room (21 °C, humidity 50%).

### 2.10. Safety Assessment of Long-Term Intake of WLE

Safety assessment was carried out at weeks 0, 4, 8, 12, and 16, including physical and vital sign measurements, hematological blood-chemical tests, and urine tests. Physical and vital sign measurements included weight, height, body mass index (BMI), systolic blood pressure (SBP), diastolic blood pressure (DBP), and heart rate. SBP, DBP, and heart rate were taken on the left arm under standard, quiet conditions after at least 5 min of rest. The blood biochemical tests measured aspartate aminotransferase (AST), alanine aminotransferase (ALT), gamma-glutamyl transpeptidase (γ-GT), lactate dehydrogenase (LDH), total protein, albumin/globulin ratio, triglycerides, total cholesterol, HDL-cholesterol, LDL-cholesterol, free fatty acids, blood urea nitrogen (BUN), uric acid, creatinine, sodium (Na^+^), potassium (K^+^), calcium (Ca^2+^), chloride (Cl^−^), inorganic phosphorus, magnesium (Mg^2+^), iron (Fe^2+^), blood glucose, hemoglobin A1c (HbA1c), alkaline phosphorus, total ketone bodies, acetoacetic acid, 3-hydroxybutyric acid, and lipoprotein. In the hematological tests, white blood cells, red blood cells, hemoglobin, hematocrit, mean corpuscular volume (MCV), mean corpuscular hemoglobin level (MCH), mean corpuscular hematocrit concentration (MCHC), and platelet count were measured. The urine tests monitored pH, specific gravity, protein, sugar, urobilinogen, occult blood, bilirubin, and ketone bodies. The above tests were performed by a third-party laboratory (H.U. Frontier Co., Ltd., Osaka, Japan). Adverse events during the study were also monitored by interviews by physician and self-reports.

### 2.11. Statistical Analysis

The safety parameter data are presented as mean ± standard deviation (SD). Comparisons between multiple groups were performed by two-way analysis of variance (two-way ANOVA) with post hoc analysis. The male–female ratio of each group was compared using a chi-square test. Intergroup (test vs. placebo) comparisons were assessed using an unpaired *t*-test or Welch’s *t*-test for parametric data. The Mann–Whitney U test was performed for non-parametric data. Intragroup (vs. week 0) comparison was evaluated using Dunnett’s test for parametric data or the Wilcoxon signed-rank test for non-parametric data. Fisher’s exact test was performed to compare the intergroup adverse event cases. The effect of the test supplement vs. placebo on TEWL was evaluated by using mixed-effects models with intervention, baseline, measurement sites, gender, and age as fixed variables and subject as a random effect. The comparison of water content was evaluated using mixed-effects models with intervention, measurement sites, gender, and age as fixed variables and subject as a random effect. Significance was set at *p* < 0.05. All analyses were performed with SPSS statistical software ver. 25.0 (IBM Corp., Armonk, NY, USA) or R version 4.2.1 (The R Foundation, Vienna, Austria).

## 3. Results

### 3.1. Identification of Ceramides and Glucosylceramides Derived from Wine Lees

To identify ceramides and glucosylceramides, briefly, the concentrated ethanolic extract of wine lees was partitioned between n-hexane and aqueous methanol fractions. The n-hexane fraction was further separated by chromatography on a silica gel column successively to yield ceramides and glucosylceramides. The chromatogram of ceramides and glucosylceramides analyzed in HPLC-ELSD is shown in Figure 2.

The molecular species of the isolated ceramides and glucosylceramides were determined by using LC-MS/MS and by comparing their fragmentation patterns with the reference spectra [12,22] and the LIPID MAPS Structure Database (LMSD), as shown in Table 2. Ceramides isolated from WLE were found to be composed of trihydroxy SBs, including both saturated and trans-unsaturated varieties, for instance, t18:0 and t18:1 (8t). Furthermore, not only trihydroxy SBs but also dihydroxy d18: SBs were identified in glucosylceramides derived from WLE.

### 3.2. In Vitro Evaluation of the Bioactivity of WLE

To evaluate the bioactivity of WLE-derived ceramides and glucosylceramides, NHEKs were treated with WLE, and the mRNA expression levels were assessed as shown in Figure 3. The transcription of cornified envelop formation-related genes*, PFR*, *INV,* and *LOR* was significantly upregulated by WLE treatment at 10 μg/mL. Moreover, the relative expression of *SPTLC 2, CerS 1,* and *CerS 3*, which were involved in de novo ceramide synthesis, also increased significantly in response to WLE at 10 μg/mL.

### 3.3. Effect of Ceramides and Glucosylceramides Isolated from WLE on the 3DHEE Ceramide Content

To date, stratum corneum ceramides are classified into 20 classes of free ceramides and 5 classes of protein-bound ceramides based on the combination of LCB and FA [23]. In particular, the level of ceramide in the stratum corneum, specifically non-hydroxy-sphingosine (NS), non-hydroxy-phytosphingosine (NP), and α-hydroxy-phytosphingosine (AP), has been found to be negatively correlated with TEWL [24]. Therefore, in this study, the effects of WLE and its constituent ceramides and glucosylceramides on the ceramide content of 3DHEEs were evaluated. The retardation factor (Rf) values of the ceramides in the 3DHEEs were obtained by HPTLC (Figure 4A). In accordance with the in vitro study with NHEKs, the addition of a 0.82 µg/mL mixture of ceramides and glucosylceramides, which was equivalent to 10 µg/mL WLE, significantly increased the ceramide NS content of the 3DHEEs (*n* = 4, *p* < 0.05) (Figure 4B). However, no significant increase was observed in ceramide NP and AP. Importantly, the addition of ceramides and glucosylceramides isolated from WLE significantly increased ceramide NS in the 3DHEEs to the same extent as WLE, indicating that the activity of WLE was attributed to the constituent ceramides and glucosylceramides.

### 3.4. Subject Demographics

A total of 30 eligible healthy volunteers consented to participate in the study and were randomly allocated to the test group (*n* = 15: male 8; female 7) or the placebo group (*n* = 15: male 7; female 8) (Table 3). Due to personal reasons irrelevant to the study, one participant of the test group withdrew during the intervention; as a result, 29 participants completed the study (test group *n* = 14, male 7, female 7; placebo group: *n* = 15, male 7, female 8). The flow diagram of the study is shown in Figure 5.

### 3.5. Efficacy Assessment

TEWL and water content were measured on the forearms of participants at 0 (baseline), 4, 8, and 12 weeks after the administration started. The baseline TEWL and water content showed no significant differences between the placebo and the test group (Table 3). After 4 weeks of administration, the TEWL values were 4.59 (95% CI 4.16, 5.03) for the placebo group and 5.19 (95% CI 4.70, 5.68) for the test group (Figure 6A). The test group exhibited a tendency (*p* = 0.07) toward lower TEWL (4.67, 95% CI: 4.16, 5.17) compared to the placebo group (5.19, 95% CI: 4.70, 5.68) after 8 weeks of intervention. Furthermore, after 12 weeks of intervention, significant differences (*p* = 0.04) were observed between the test (4.38, 95% CI: 3.87, 4.88) and the placebo group (5.00, 95% CI: 4.51, 5.49). Water content showed no significant difference between the placebo and test supplement during the study (Figure 6B). No significant differences were observed in the VAS of itching sensation and the Japanese version of the Skindex-29 questionnaires.

### 3.6. Safety Assessment

The safety evaluation was carried out at weeks 0, 4, 8, 12, and 16 following ingestion of the test supplement or placebo. The results of physical and vital sign measurements are shown in Table 4. No significant differences in physical or vital signs were observed between the placebo and the test group throughout the study. In biochemical and hematological analysis (Appendix A), a significant difference was observed between the groups in lipoprotein parameters since the beginning of the study period (0-week); however, both groups exhibited similar trends. None of the quantitative and qualitative urinalyses (Appendix A) were significantly different between the test group and the placebo group during the study. According to the physician’s assessment, no adverse events possibly linked to the test supplement were reported.

## 4. Discussion

Ceramides, essential components of the stratum corneum, are crucial for maintaining skin hydration and integrity [25]. A growing awareness of the diet’s role in skin health has identified the consumption of dietary ceramides as a promising, effective, and safe strategy for preserving skin barrier function. A meta-analysis of seven clinical studies on oral ceramide intake revealed a significant increase in skin hydration and a decrease in TEWL compared to the placebo [26].

In this study, we reported for the first time the isolation and identification of ceramides and glucosylceramides from WLE and the evaluation of their in vitro activity. Specifically, we examined the expression of *PFR*, *INV*, and *LOR*, which encode proteins secreted during the induction of keratinocyte differentiation, leading to cornified cell envelope formation. We observed a significant increase in the transcription of these genes induced by WLE. Moreover, the increased expression of genes involved in keratinocyte differentiation is suggested to be one of the underlying skin barrier-enhancing mechanisms of ceramides [27,28]. In addition, we found that the ethanolic extract of wine lees upregulated the expression of multiple genes (*SPTLC 2*, *CerS 1,* and *CerS 3*) that encode essential enzymes involved in the de novo synthesis of ceramides in epidermal keratinocytes. *SPTLC 2* encodes serine palmitoyltransferase, which catalyzes the condensation of serine and palmitoyl-CoA at the initial stage of de novo ceramide synthesis. Being encoded by *CerS*, ceramide synthases play a pivotal role in the N-acyltranslation of fatty acids, leading to the formation of the basic ceramide structure [29]. This finding is supported by the observation of increased ceramide NS content in 3DHEEs treated with WLE or mixtures of ceramides and glucosylceramides isolated from wine lees. Furthermore, the level of ceramide NS in the stratum corneum is known to correlate with TEWL [24]. Importantly, the mixtures of ceramides and glucosylceramides increased the ceramide NS content of the 3DHEEs to the same extent as equivalent amounts of WLE, suggesting that the activity of WLE is primarily attributable to its constituent ceramides and glucosylceramides. Taken together, these results suggest that the WLE and WLE-derived ceramides and glucosylceramides enhance skin barrier function through the dual mechanisms of the enhancement of cornified envelope formation and ceramide synthesis.

In a preliminary study, WLE was assessed in a single-dose toxicity test on mice at 2000 mg/kg (equivalent to 9800 mg/kg for a 60 kg human adult using allometric scaling [30]), and no adverse events were observed. Following this, we conducted a short-term human clinical study in Japan from September to October 2022. This study aimed to evaluate the effectiveness of WLE-derived ceramides and glucosylceramides. Participants ingested daily doses of the test supplement (100 mg/day and 500 mg/day) containing WLE-derived ceramides and glucosylceramides (2 mg and 10 mg, respectively) for 4 weeks, and TEWL on the forearm was significantly reduced. Based on these findings, we further investigated the effectiveness and safety of oral administration of the test supplement containing 100 mg/day of WLE-derived ceramides and glucosylceramides (2 mg) over an extended period (12 weeks). This was performed through a randomized, double-blind, and placebo-controlled study with healthy individuals. The results demonstrated that the oral administration of WLE-derived ceramides and glucosylceramides for 12 weeks significantly lowered TEWL. Remarkably, the time series analysis of TEWL indicated a progressive increase in the difference between the test and placebo groups with the duration of administration. However, no significant difference was observed in water content among participants. It is important to note that all participants in this study were healthy adults without major skin issues. Future research should explore the effects of WLE-derived ceramides and glucosylceramides on water content in individuals with more severe skin conditions. Regarding the safety assessments, since no adverse events or significant deviation from the normal values of safety parameters were reported or observed throughout the study period, this suggests that WLE-derived ceramides and glucosylceramides can be used as a safe source of ceramides for the enhancement of skin barrier function.

However, this study has several limitations that should be acknowledged. Firstly, the sample size was modest, potentially limiting the generalizability of the findings. Secondly, the measurements were performed only on the forearms. Another limitation is the duration of 12 weeks for the study period, which may not capture long-term effects associated with the oral administration of WLE-derived ceramides and glucosylceramides. Therefore, future investigations are required to examine different anatomic sites with a larger sample size and longer study period.

The oral bioavailability of ceramides has been intensively studied in animal models. The distribution of ^3^H-ceramides across various body parts in rats, including epidermis, following a single administration and gastrointestinal absorption has been reported [31]. Glucosylceramides are thought to be metabolized in the gastrointestinal tract to form sphingoid bases, thereby reaching the skin, where the sphingoid bases are resynthesized locally to ceramides and glucosylceramides to exert their effects [15]. On the other hand, recent studies have demonstrated that dietary ceramides and glucosylceramides could also be directly absorbed into the blood circulation without undergoing digestion. Additionally, a portion of dietary glucosylceramides could be potentially hydrolyzed and absorbed as ceramide molecules after digestion [12]. Hence, it could be inferred that orally ingested WLE-derived ceramides and glucosylceramides may follow a metabolic pathway analogous to that described in animal studies.

## 5. Conclusions

In this study, we demonstrated that the oral ingestion of WLE-derived ceramides and glucosylceramides for 12 weeks improved the skin barrier function of healthy individuals. The in vitro study suggested that the WLE-derived ceramides and glucosylceramides and enhanced cornified envelope formation and increased epidermal ceramide content. No adverse events or significant deviation from the normal values of safety parameters were reported or observed throughout the study period. These results collectively underline the safety and effectiveness of WLE-derived ceramides and glucosylceramides as bioactives for enhancing skin barrier function.

## Figures and Tables

**Figure 1 nutrients-16-02100-f001:**
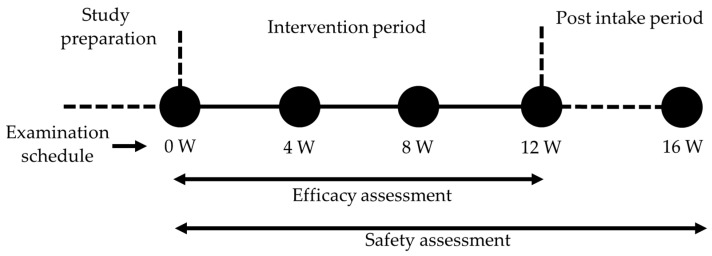
Study schedule. Efficacy assessment was conducted at weeks 0, 4, 8, and 12. Safety assessment was performed at weeks 0, 4, 8, 12, and 16.

**Figure 2 nutrients-16-02100-f002:**
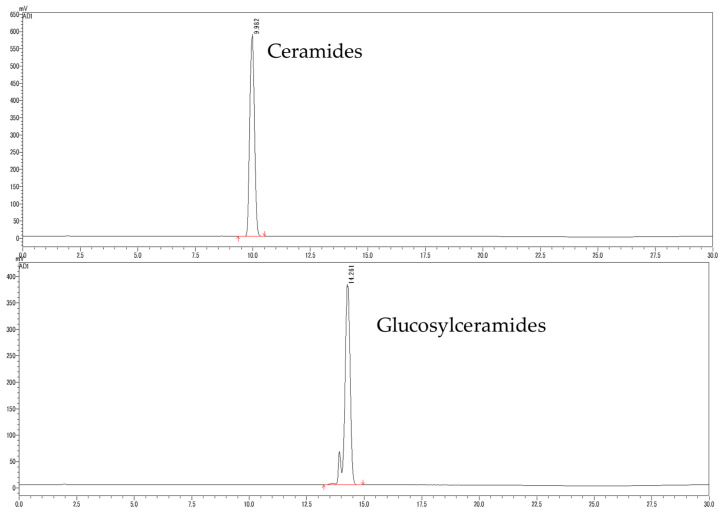
Chromatogram of ceramides and glucosylceramides isolated from WLE.

**Figure 3 nutrients-16-02100-f003:**
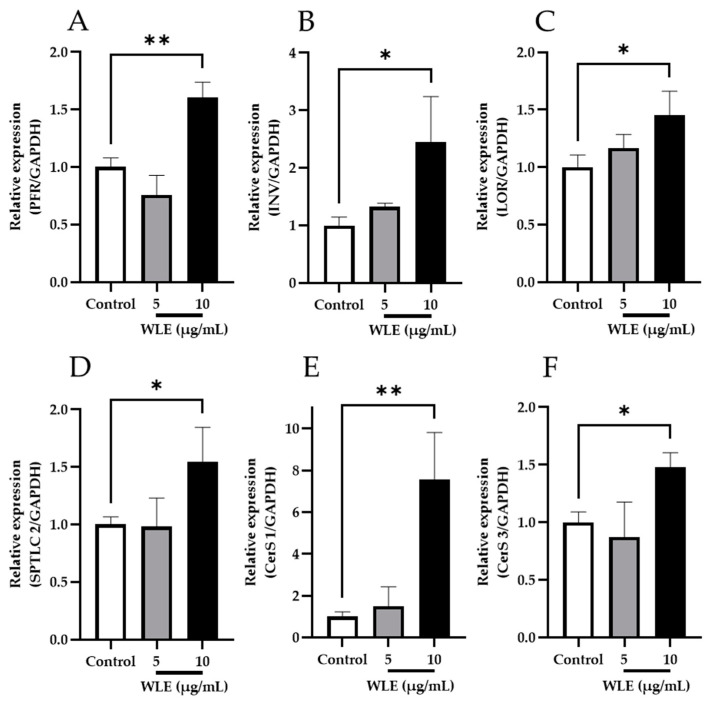
Effects of WLE on the mRNA expression levels of de novo ceramide synthesis-related enzymes (**A**–**C**) and cornified envelope (CE) formation (**D**–**F**). The results indicated are mean ± SD. Significant differences were examined by one-way analysis of variance (ANOVA) followed by Dunnett’s test (*n* = 3, * *p* < 0.05; ** *p* < 0.01). (**A**) *PFR*; (**B**) *INV*; (**C**) *LOR* (**D**) *SPTLC 2*; (**E**) *CerS 1*; (**F**) *CerS 3*.

**Figure 4 nutrients-16-02100-f004:**
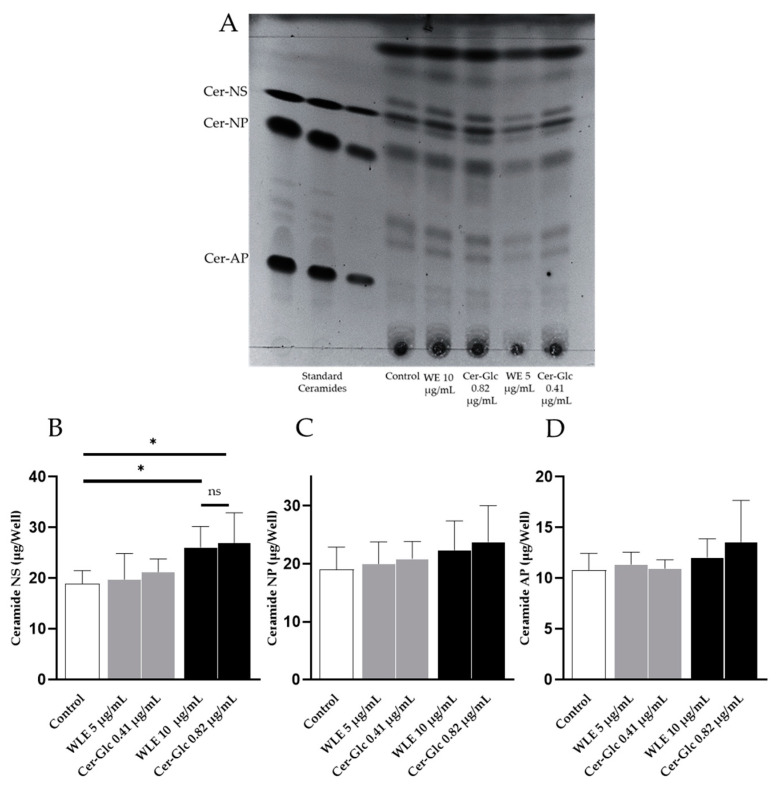
Effects of WLE and the mixtures of ceramides and glucosylceramides on the ceramide content of the 3DHEEs were measured using HPTLC. (**A**) HPTLC image of lipids from 3DHEEs compared with commercially available standard ceramides. (**B**–**D**) Ceramide content in 3DHEEs after the application of WLE and mixtures of ceramides and glucosylceramides isolated from WLE. Results are mean ± SD. Statistical significance of differences was determined by Dunnett’s test (*n* = 4, * *p* < 0.05).

**Figure 5 nutrients-16-02100-f005:**
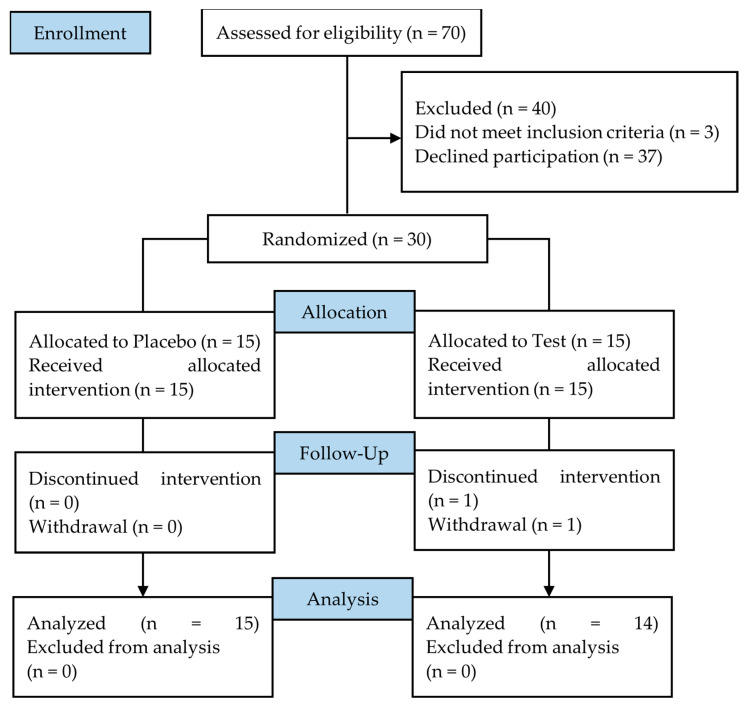
Flow diagram of the clinical study.

**Figure 6 nutrients-16-02100-f006:**
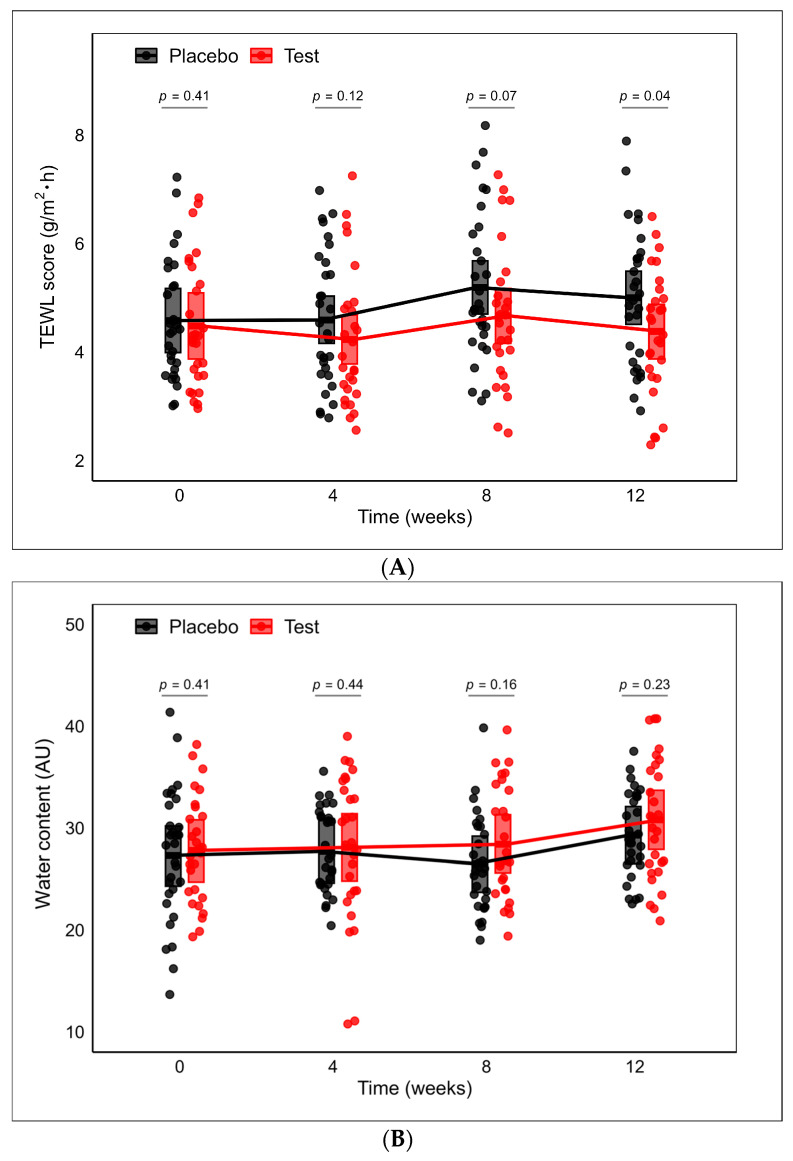
(**A**) The trajectories of TEWL scores over time. Between-group comparison at each time point was conducted using a mixed-effects model with treatment, TEWL at baseline, gender, age, and arm of data collection as fixed variables and subject as a random effect. (**B**) The trajectories of water content over time. Between-group comparison at each time point was conducted using a mixed-effects model with treatment, gender, age, and arm of data collection as fixed variables and subject as a random effect. The middle lines of the crossbars denote the least squares (LS) means. The upper and lower edges of the crossbars indicate the upper and lower limits of the 95% confidence intervals (CIs), respectively.

**Table 1 nutrients-16-02100-t001:** Nutritional composition of the test supplement and placebo.

	Nutritional Value per Serving (100 mg)	
	Test Supplement	Placebo
Carbohydrate (g)	0.069	0.095
Protein (g)	0.002	0.000
Fat (g)	0.024	0.000
Water (g)	0.004	0.000
Sodium (mg)	0.020	0.000
Ash * (g)	0.002	0.050
Energy (kcal)	0.496	0.381
Total Ceramides (mg)	2.000	0.000

* Ash refers to the inorganics remaining after complete removal of moisture, volatiles, and organic matter by heating.

**Table 2 nutrients-16-02100-t002:** Molecular species of ceramides and glucosylceramides isolated from WLE.

Ceramide Molecular Species	Precursor Ion (Q1) *m*/*z*	Product Ion (Q3) *m*/*z*	Glucosyl- Ceramide Molecular Species	Precursor Ion (Q1) *m*/*z*	Product Ion (Q3) *m*/*z*
	[M+H]⁺			[M+H]⁺	
t18:1/h22:0	654.6	262.3, 280.3, 298.3	d18:2/c16:1	696.7	262.3
t18:0/h22:0	656.7	264.3, 282.3, 300.3	d18:2/h16:0	714.5	262.3
t18:1/h23:0	668.6	262.3, 280.3, 298.3	d19:2/h18:1	737.5	276.3
t18:1/h24:0	682.7	262.3, 280.3, 298.3	d18:2/c20:1	752.5	262.3
t18:0/h24:0	684.7	264.3, 282.3, 300.3	t18:1/h22:0	816.6	262.3, 280.3
t18:1/h25:0	696.7	262.3, 280.3, 298.3	d18:1/c26:1	838.6	264.3
t18:1/h26:1	710.7	262.3, 280.3, 298.3	t18:1/h24:0	844.6	262.3, 280.3

**Table 3 nutrients-16-02100-t003:** Baseline subject demographics.

	Normal Range	Placebo (*n* = 15)			Test (*n* = 15)			Test vs. Placebo
		Mean	±	SD	Mean	±	SD	*p*-Value
Age (years)		38.7	±	9.8	37.3	±	8.4	0.702
Sex (M/F)		7	/	8	8	/	7	1
Height (cm)		163.3	±	8.1	168	±	5.8	0.146
Weight (kg)		62.7	±	10.6	64.7	±	11.2	0.633
Body Mass Index (BMI) (kg/m^2^)	18.5–24.9	23.4	±	2.3	22.8	±	3.2	0.596
SBP (mmHg)	−139	121.3	±	12.1	121.9	±	12.7	0.899
DBP (mmHg)	−89	73.7	±	9.0	77.3	±	9.6	0.316
Heart rate(bpm)	60–99	72.3	±	10.8	68.9	±	7.0	0.322
TEWL(g/m²·h)		4.6	±	1.0	4.5	±	1.1	0.857
Skin Hydration (%)		27.4	±	6.1	27.6	±	4.9	0.927

Values are mean ± standard deviation (SD). Abbreviations: SBP, systolic blood pressure; DBP, diastolic blood pressure; TEWL, transepidermal water loss.

**Table 4 nutrients-16-02100-t004:** Physical and vital sign parameters during the study.

	Normal Range	Group	0 w	4 w	8 w	12 w	16 w	
Height (cm)		Placebo	163.3 ± 8.1					
		Test	167.4 ± 5.5					
Weight (kg)		Placebo	62.7 ± 10.6	62.4 ± 10.3	62.3 ± 10.4	62.1 ± 10.4	62.4 ± 10.6	
		Test	64.3 ± 11.5	64 ± 10.8	63.9 ± 10.7	63.8 ± 10.8	62.9 ± 11.4	
BMI (kg/m^2^)	18.5–24.9	Placebo	23.4 ± 2.3	23.3 ± 2.3	23.2 ± 2.3	23.2 ± 2.3	23.2 ± 2.4	
		Test	22.8 ± 3.3	22.7 ± 3.1	22.7 ± 3.1	22.6 ± 3.1	22.4 ± 3.4	
SBP (mmHg)	−139	Placebo	121.3 ± 12.1	121.5 ± 12.4	119.3 ± 11.0	117.7 ± 8.5	117.1 ± 12.1	
		Test	120.4 ± 11.8	117.5 ± 9.2	121.1 ± 10.6	119.16 ± 11.2	118.8 ± 11.0	
DBP (mmHg)	−89	Placebo	73.7 ± 9.0	75.1 ± 8.9	74.8 ± 8.4	73.7 ± 7.0	71.3 ± 10.4	
		Test	76.1 ± 8.7	77 ± 9.0	75.8 ± 7.3	73.0 ± 6.4	74.0 ± 8.6	
Heart rate (bpm)	60–99	Placebo	72.3 ± 10.8	68.6 ± 7.8	72.3 ± 8.5	70.9 ± 11.7	71.9 ± 9.8	
		Test	68.5 ± 7.1	70.6 ± 9.6	70.4 ± 8.5	71.5 ± 8.4	77.6 ± 7.7	#

Values are presented as mean ± SD. Dunnett’s test was used for intragroup comparison with 0-week # *p* < 0.05.

## Data Availability

Due to ethical concerns regarding the clinical study, the dataset generated during and analyzed during the present study is not publicly available. De-identified participant data are, however, available from the corresponding author upon reasonable request.

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
