# Peer review of "Efficacy and Safety of Oral Administration of Wine Lees Extract (WLE)-Derived Ceramides and Glucosylceramides in Enhancing Skin Barrier Function: A Randomized, Double-Blind, Placebo-Controlled Study"

_nutrients, 2024, doi:10.3390/nu16132100_

Round 1

Reviewer 1 Report

Comments and Suggestions for Authors

I find this study on wine extract and its efficacy in skin barrier function very interesting and the manuscript is well prepared. I believe this manuscript will add to the body of literature in the cosmetic science field as barrier function is a very important topic not only in skin health but also in the skin aging field. References used for the manuscript seem appropriate and conclusions are based on the presented results. The methodology section is written in detail and the study could easily be reproduced.

I only have minor suggestions for the authors:

1. write an abstract according to the CONSORT checklist for abstracts

2. add a limitation section to the discussion

3. Improve English language and style

4. improve quality of figure 4 and 6

Comments on the Quality of English Language

minor improvements needed

Reviewer 2 Report

Comments and Suggestions for Authors

Dear Authors,

The manuscript entitled "Efficacy and Safety of Oral Administration of Wine Lees Extract (WLE)-Derived Ceramides and Glucosylceramides in Enhancing Skin Barrier Function: A Randomized, Double-Blind, Placebo-Controlled Study" is a well designed study and in my opinion it will have an impact to the readers in the field. The statistics have been well performed and the whole study in general diserves to proceed to the next step of the publication process. Only minor issues i have to report for the submiited study.

Could the authors reduce the size of the of title

In the discussion section, the first paragraph is too long and needs to be reduced or even be removed.

Also, please check the whole manuscript for zny grammar or phase errors
